Thumbs up: movements made by the thumb are smoother and larger than fingers in finger-thumb opposition tasks

Rachaveti Dhanush
Chakrabhavi Niranjan
Shankar Vaisakh
SKM Varadhan skm@iitm.ac.in
Department of Applied Mechanics, Indian Institute of Technology Madras , Chennai , Tamil Nadu , India
Keogh Justin
Electronic publication date: 2018 Oct 18
Publication date: 2018
Volume: 6
Electronic Location ID: e5763
Received 2018 Apr 24; Accepted 2018 Sep 17
Copyright: ©2018 Rachaveti et al.
Copyright year: 2018
Copyright holder: Rachaveti et al.
License: This is an open access article distributed under the terms of the Creative Commons Attribution License, which permits unrestricted use, distribution, reproduction and adaptation in any medium and for any purpose provided that it is properly attributed. For attribution, the original author(s), title, publication source (PeerJ) and either DOI or URL of the article must be cited.
License URL: https://creativecommons.org/licenses/by/4.0/

Keywords: Thumb, Kinematics, Smoothness, Thumb-Finger Opposition Movements

Funding: Department of Science & Technology SR/CSRI/97/2014 Cognitive Science Research initiative (CSRI) This work was financially supported by Department of Science & Technology, Government of India, vide Reference No SR/CSRI/97/2014 under Cognitive Science Research initiative (CSRI) to carry out this work (awarded to Varadhan SKM). The funders had no role in study design, data collection and analysis, decision to publish, or preparation of the manuscript.

==============================
Background

In humans, the thumb plays a crucial role in producing finger opposition movements. These movements form the basis of several activities of the hand. Hence these movements have been used to study phenomena like prehension, motor control, motor learning, etc. Although such tasks have been studied extensively, the relative contribution of the thumb vis-à-vis the fingers in finger opposition tasks is not well understood. In this study, we investigated the kinematics of thumb and fingers in a simple finger opposition task. Further, we quantified the relative contribution and the movement smoothness aspects and compared these between fingers and thumb.

Methods

Eight, young healthy participants (four males and four females) were asked to perform a full finger to thumb opposition movement, where they were required to reach for different phalanges of the fingers. Position (X, Y and Z) of individual segments of the four fingers and the thumb were measured with reference to the wrist by a 16-sensor kinematics measurement system. Displacements and velocities were computed. An index, displacement ratio, that quantifies the relative contribution of thumb and fingers was computed from displacement data. Velocity data was used to quantify the smoothness of movement of thumb and fingers.

Results

The Displacement Ratio showed that contribution of the thumb is higher than contribution of any other target finger or target phalanges, except for the distal phalanx of the index and middle fingers. Smoothness of movement of the thumb was higher than all the finger phalanges in all cases.

Conclusion

We conclude that in the task considered (thumb opposition movements to different targets within the hand & fingers), the thumb made a greater relative contribution in terms of displacement ratio and also produced smoother movements. However, smoothness of thumb did not vary depending on the target. This suggests that the traditional notion of the thumb being a special digit when compared to other fingers is true at least for the opposition movements considered in this study.

Introduction

The evolution of opposable thumb in humans is considered to be a crucial feature in the development of fine and dexterous hand movements (Napier, 1956). The opposable thumb can be considered to uniquely distinguish humans from other animals, including non-human primates. Thumbs in non-human primates are diminutive with the absence of complete opposition movements. It has been reported that thumbs in non-human primates are often used for clubbing and throwing stones (Young, 2003). So, it is possible to argue that the evolution of thumb in humans has enabled them to produce full finger opposition movements. This probably helps humans perform daily activities that require fine and precise control (Signori et al., 2017). Such activities include gripping of a spoon (precision grip), holding a bottle (power grip), etc. Because of all these reasons, studies of finger opposition movements are frequently used in addressing problems related to central nervous system (CNS) strategy in prehension, cognition, motor control and motor learning.

Several studies in the field of prehension have focused on understanding how finger forces are controlled and how the forces are coordinated with each other to perform the act of grasping. Modalities such as a change in equilibrium of the object, prevention of slip etc., are among some of the approaches used in these studies (Zatsiorsky & Latash, 2004; Zatsiorsky & Latash, 2008; Pataky et al., 2013; Martin, Latash & Zatsiorsky, 2011; Park et al., 2012; Slota, Latash & Zatsiorsky, 2012). One study addressed the question of how the thumb and the other fingers produce both opposition movements and parallel movements in a five-finger pressing task (Olafsdottir, Zatsiorsky & Latash, 2005). The main result from that study was that the thumb could be considered as a fifth finger, at least from the viewpoint of multi-finger interaction. Using kinematics, several studies have documented the changes in hand postures based on the shape and contours of objects (Santello & Soechting, 2000; Santello, Flanders & Soechting, 1998). A different study has demonstrated that there were task (object) specific changes in hand shaping before grasping (Santello & Soechting, 1998). These studies have focused on overall hand function rather than thumb–finger opposition.

In the field of cognitive learning, several studies have used finger opposition movements to investigate the role of phenomena such as memory consolidation (Karni et al., 1995; Karni, 1996), representations (Rozanov, Keren & Karni, 2010) and sleep patterns (Doyon et al., 2009) in learning. Some of these studies focused on the timing or temporal task outcomes such as reaction time, movement time, etc. (Friedman & Korman, 2012; Friedman & Korman, 2016). In the field of motor control and learning, some studies have reported the role of finger movements in studying the phenomenon of co-articulation (fluency). These studies were on American sign language (Jerde, Soechting & Flanders, 2003), handwriting (Wright, 1993), piano playing (Winges et al., 2013) and typing (Flanders & Soechting, 1992) where context specific learning (co-articulation) was quantified. One study (Signori et al., 2017) has attempted to quantify the finger opposition movements as an assessment of health in participants by calculating the movement rate, touch time when thumb tip opposes fingertips.

Within the field of motor learning, it has been observed that characteristics of movements change with practice—both in spatial and temporal domains (Sosnik et al., 2004). A hallmark of well-trained movement is that, their trajectories are continuous, smooth (Balasubramanian, Melendez-Calderon & Burdet, 2012; Krylow & Rymer, 1997) and has minimal jerk (Flash & Hogan, 1985; Hogan & Sternad, 2009). These features have also been used to distinguish novices from experts (Karni et al., 1995; Karni, 1996; Karni et al., 1998). Hence, although finger opposition movements have been used as a tool to study different phenomena in different areas of research, the question of how exactly the thumb moves to different finger phalanges remains unexplored. We believe that it is essential to investigate the characteristics of movements made by the thumb to reach different target positions on hand, i.e., on all digits (Index, Middle, Ring and Little) and on all phalanges (Distal, middle and proximal). Reaching to different finger phalanges/joints is used in some cultures to count. For example, in India, several communities teach children to count using the MCP/PIP/DIP joints of various fingers representing different digits (numbers). In this counting approach, the joints in right hand represent units (×100) and the joints in left hand represent tens (×101). Note that, in this approach, the PIP of middle and ring fingers are not used to count. A similar approach is used in counting tones/notes for traditional south Indian music (Carnatic music).

In the current study, we examine the relative contribution of the fingers to the opposition movement of thumb. Further, we performed detailed analysis to understand how movement smoothness varies in the fingers and thumb during the opposition movements. Two hypotheses were framed and tested. The first hypothesis was that the thumb and fingers would contribute approximately equally when trying to reach a distant target (such as the distal phalanx of the little finger) but the contribution of thumb would be higher in reaching a nearer target (such as the distal phalanx of the index finger). The second Hypothesis was that between the two effectors that reach for each other (say a finger phalanx and thumb), movements of the effector that contributes more would be smoother than movements of the effector that contributes less. i.e., if the thumb’s relative contribution when compared with the finger is greater, then the thumb will also produce smoother movements.

Materials and Methods

Participants

Eight healthy, right hand dominant participants (four males & four females (mean ± standard deviation Age: 24.8 ± 2.6 yrs., Height: 166.5 ± 7.0 cm, Weight: 63.2 ± 9.5 kg)) volunteered for the study. The participants had no history of any neuro-motor disorder or trauma to the hand or fingers. Their handedness was determined using the Edinburgh handedness inventory (Oldfield, 1971). All participants provided written informed consent before participating in the experiment. All experimental procedures were approved by institutional ethics committee of IIT Madras (IEC/2018/01/SKM-2/03)

Experimental setup

In this experiment, participants were instructed to place their hand supinated on a height adjustable chair—table setup with fingers abducted comfortably in the rest position as seen in Fig. 1. An electromagnetic tracking system (Liberty; Polhemus, Inc., Colchester, VT, USA.) consisting of 16 micro sensors were used to collect kinematic information (X, Y, Z,) of the hand.

Figure 1 Schematic diagram showing the experimental setup and sensor placement.

(A) Participants placed their hand on a height adjustable chair/table setup in a comfortable supinated position during the experiment. Participants were shown a user interface as shown above, where the segments (phalanx and thumb) to be moved was highlighted in black color. (B) Sensors were placed on the dorsal side of the participant’s hand on each of the finger phalanges as well as on thumb segments. (C) Photograph shows the first author performing the experimental task for illustration purpose. Photograph by Vaisakh Shankar.

This complete 16 sensor system provided one sensor for each phalanx on fingers and thumb (5 × 3) and one reference sensor (15 + 1 = 16 sensors). These sensors have a diameter of 1.8 mm, resolution of 1.27 µm and 0.0004 deg, static accuracy of 0.76 mm for position and 0.15 deg for orientation. All of these sensors were placed on the dorsal surface of the fingers and thumb. The reference sensor was placed on the dorsal wrist.

A customized LabVIEW program was written for user interface and data was collected at 100 Hz. A standard 4 –inch source transmitter (Polhemus, Inc.) was used to define the lab co-ordinate system. The participant’s hand was placed on the wooden table approximately at X = 9 cm, Y = 6 cm, Z = 11 cm from the transmitter throughout the experiment. Figure 1 illustrates the experimental setup.

Experimental task

Participants were instructed to make an opposition movement with the tip of the thumb and finger phalanx moving towards each other. Such thumb to phalanx opposition movements are used in some cultures to count and in approaches involving hand (or glove) based text input to computers. In this experiment, participants were required reach to the target finger phalanx by making thumb-finger opposition movement of their hand. The target phalanx was highlighted on the screen. They were asked to start the movement once they hear a beep sound. They were asked to perform this task at a comfortable speed.

Experimental protocol

Movement of thumb towards a specific finger phalanx constituted a task, resulting in 12 tasks (four fingers × three phalanges). Each task was repeated 15 times, each repetition was considered as a trial. The tasks were presented to the participants in a random fashion (block randomized i.e., task was not selected essentially in a sequential manner from index to little finger or from distal to proximal phalanx but randomly), from the pool of 12 tasks. Each trial lasted 3 secs. Each participant performed 180 trials (12 targets × 15 trials). One of the experimenters monitored the movement of the thumb and the fingers, if the thumb did not touch the instructed phalanx, trial was repeated. In total, about 4% of the trials were repeated. Each trial consisted of movements of effectors (thumb and finger phalanges) toward each other (onward movements) and movements away from each other. For the purpose of this study, only onward movements were analysed further.

Data analysis

Raw data was filtered at 5 Hz using by using second order, zero-lag low-pass Butterworth filter. Filtered data was used to calculate the resultant displacement RD, velocity RV was computed using five-point stencil method (Abramowitz & Stegun, 1965). These measurements were made with respect to hand coordinate frame (i.e., with respect to reference sensor placed on the wrist). Separate calculation was performed for thumb and a finger phalanx in a given task. For example, in task 1, RD and RV was calculated separately for thumb and distal phalanx of index finger. RD=x−x02+y−y02+z−z02

Where x0, y0 and z0 are initial displacement measurements in rest position. RV=dx−x0dt2+dy−y0dt2+dz−z0dt2

Onset of movement was determined as the first time when the 5% of maximum velocity was crossed. Traversing back in time in velocity profile from peak velocity, we determine the time point at which velocity becomes less than 5% of the peak velocity. The successive time point (forward in time) was denoted as the onset of movement. End of the movement was determined as the first-time when the velocity goes below 5% of maximum velocity. This value was determined by traversing forward in time from the point of peak velocity and the last point in the velocity profile in which velocity remains above 5% of the peak velocity was considered as end of movement. In the few cases in which this algorithm detected a spurious movement, a manual method combined with a slightly different algorithm was used to identify the onset and end of movement (please see the algorithm attached as supplementary material). We computed thumb and finger displacements as follows

thumbdisplacement=RDTend of movement−RDTonset of movement

fingerdisplacement=RDFend of movement−RDFonset of movement

where RDT and RDF are resultant displacement of thumb and finger.

Displacement ratio (D)

The amount of displacement produced by finger phalanges and thumb in an opposition movement was considered for further analysis. The ratio of absolute value of thumb displacement to the total absolute displacement of thumb and the finger phalanx was defined as Displacement ratio of thumb (DT). Similarly, the ratio of absolute displacement of finger phalanx to the total absolute displacement of thumb and finger was termed as Displacement ratio of the Finger phalanx (DF).

DT=thumbdisplacementthumbdisplacement+fingerdisplacement

DF=fingerdisplacementthumbdisplacement+fingerdisplacement.

Both these indices, were limited by 0 to +1. Therefore, for statistical comparison, fisher’s Z-transform was performed on these indices.

ZD_T=0.5 ∗ ln1+DT1−DT

ZD_F=0.5 ∗ ln1+DF1−DF.

Smoothness measure

The Velocity profile obtained from onward movement segment was used to determine the movement smoothness of each finger segment and thumb distal segment. For this, a method discussed in Balasubramanian, Melendez-Calderon & Burdet (2012) was used, where velocity profile of the movement was transformed to Fourier magnitude spectrum. Then the arc length (length along a curve) was calculated for the amplitude and frequency normalized magnitude spectrum. The negative value of this arc length was termed as spectral arc length (SAL) and it reduces (lesser negative value) with increase in movement smoothness or vice-versa.

Statistics

Two-way repeated measures ANOVA was performed with target (instructed) fingers (4 Levels: Index, Middle, Ring and Little) × target (instructed) phalanx (3 Levels: Distal, middle and proximal phalanx) as factors for displacement and smoothness measure. One-way repeated measures ANOVA was performed on displacement ratio data separately for each task (twelve such ANOVAs). Each of these ANOVA had factor moving effector (2 levels: Thumb and finger phalanx). In all cases, data were checked for violations of sphericity and the Huynh-Feldt (H-F) criterion was used adjust the number of degrees of freedom where required.

Results

Task performance

Participants moved target finger phalanx and thumb towards each other until they touched. The displacement and velocity profiles of the thumb and finger phalanges are presented in Figs. 2 and 3. The velocity profile of target finger phalanx had more undulations in the profile when compared to the velocity profile of the thumb.

Figure 2 Displacement profile of thumb and finger.

Representative data from a single participant showing displacement profile for movement of thumb towards target finger phalanx. Data was segregated using onset and end of movement times and was interpolated to 100 time points. Each section from (A–L) represents different target finger and phalanx positions. Dashed lines represent thumb displacement, while solid lines represent finger displacement. (A) Index finger: distal phalanx; (B) index finger: middle phalanx; (C) index finger: proximal phalanx; (D) middle finger: distal phalanx; (E) middle finger: middle phalanx; (F) middle finger: proximal phalanx; (G) ring finger: distal phalanx; (H) ring finger: middle phalanx; (I) ring finger: proximal phalanx; (J) little finger: distal phalanx; (K) little finger: middle phalanx; (L) little finger, proximal phalanx.

Figure 3 Velocity profile of thumb and the finger.

Representative data from a single participant, showing velocity profile for movement of thumb towards target finger phalanx. Data was segregated using onset and end of movement times and was interpolated to 100 time points. Dashed lines represent thumb velocity, while solid lines represent finger velocity. (A) Index finger: distal phalanx; (B) index finger: middle phalanx; (C) index finger: proximal phalanx; (D) middle finger: distal phalanx; (E) middle finger: middle phalanx; (F) middle finger: proximal phalanx; (G) ring finger: distal phalanx; (H) ring finger: middle phalanx; (I) ring finger: proximal phalanx; (J) little finger: distal phalanx; (K) little finger: middle phalanx; (L) little finger, proximal phalanx.

Effect of target fingers on DF and DT

The effects of target fingers on ZD_F (F(3.57,24.99) = 37.17; p < 0.001) and on ZD_T (F(2.58,18.06) = 20.60; p < 0.001) were significant according to two way repeated measures ANOVA. Post-hoc pairwise comparisons on ZD_F (DF) (mean of z-transformed ratio (mean of actual ratio)) for target fingers showed that there was significant decrease (p < 0.01), when the position of the target finger was considered to change in the order from index (0.37(0.34)) to little finger (0.21(0.20)). Similarly, post-hoc test on ZD_T (DT) (mean of z-transformed ratio (mean of actual ratio)) for target fingers showed that there was significant increase (p < 0.01), when the position of the target finger was considered to change in the order from index (0.87(0.65)) to little finger (1.23(0.79)). These findings are illustrated in Fig. 4.

Figure 4 Change of Displacement Ratio for different target finger positions.

DF reduces significantly (p < 0.01) when the target finger was changed from index to little finger while DT increases significantly (p < 0.01) when the target finger was changed in the same order. A higher value of D indicates a greater contribution. The columns and bars indicate means and standard error of means.

Figure 5 Change of Displacement Ratio for different target Phalanges positions.

DF reduces significantly (p < 0.001) when the target Phalanx was changed from distal to proximal phalanx while DT increases significantly (p < 0.001) when the target phalanx was changed in the same order. A higher value of D indicates a greater contribution. The columns and bars indicate means and standard error of means.

Effect of target phalanges on DF and DT

The effects of target phalanges on ZD_F (F(1.22,8.54) = 61.01; p < 0.001) and on ZD_T (F(1.12,7.84) = 69.79; p < 0.001) were significant according to two-way repeated measures ANOVA. Post-hoc pairwise comparisons showed significant difference (p < 0.001) between all three pairs considered (distal vs middle, middle vs proximal, proximal vs distal) for target phalanges on both indices ZD_F and ZD_T. On an average ZD_F (DF) (mean of z-transformed ratio (mean of actual ratio)) for distal, middle and proximal phalanges were 0.49(0.45), 0.32 (0.31) and 0.11 (0.11) respectively. Similarly, means of ZD_T (DT) (mean of z-transformed ratio (mean of actual ratio)) for distal, middle and proximal phalanges were 0.62 (0.54), 0.89 (0.68) and 1.54(0.88), respectively. This shows that as DF reduced, DT increased when target position was considered in the order from distal to proximal phalanx. These findings are illustrated in Fig. 5. For both DF and DT, there was no significance between the interactions of factors target fingers and phalanges.

Effect of task level differences on DF and DT

ZD_T and ZD_F were analysed together for different tasks (movement of thumb towards different phalanges) by running different twelve one-way repeated measures ANOVAs separately, one for each target. Post-hoc tests showed that only for the tasks involving distal phalanges of index and middle fingers, ZD_F showed no significance (p > 0.05) when compared with ZD_T for factor moving effector (levels—2 (finger and thumb)). In all other tasks ZD_T was significantly greater than ZD_F (p < 0.001). These findings are illustrated in Fig. 6.

Figure 6 Change of Displacement Ratio for different task.

The x-axis indicates different target positions. I, M, R, and L denote the Index, Middle, Ring and Little fingers, respectively. Similarly, P, M, D denote Proximal, Middle and Distal phalanges respectively. DT was significantly (p < 0.001) greater than DF in all the tasks except in the task involving the distal phalanx of the index (ID) and middle (MD) fingers. There was no significant difference between DT and DF in these tasks (ID and MD). The columns and bars indicate means and standard error of means.

Figure 7 Change of smoothness of phalanx and thumb for different target fingers.

The smoothness of the movement of fingers and thumb was determined using spectral arc length method, when the spectral arc length value was large, smoothness of the movement was less. The smoothness of the phalanx movement reduced significantly (p < 0.01), when target position was changed from index to little finger. There was no significant difference of smoothness of thumb movement. When target position was changed. The columns and bars indicate means and standard error of means.

Effect of target fingers on movement smoothness of phalanx and thumb

The effects of target fingers (F(2.7,18.9) = 4.15; p < 0.01) on smoothness of finger phalanx movement was significant but the same factor did not show any significant difference (p > 0.05) on smoothness of thumb movement according to two-way repeated measures ANOVA. Post-hoc pairwise comparisons on smoothness of phalanx movement showed significant decrease (p < 0.01), when the position of the target finger was considered to change in the order from index (−1.76) to little finger (−1.85). These finding are seen in the Fig. 7.

Effects of target phalanges on movement smoothness of phalanx and thumb

The effects of target phalanges (F(1.12,7.84) = 49.94; p < 0.001) on smoothness of phalanx movement were significant according to two way repeated measures ANOVA, but similar effect was not seen on smoothness of thumb movement. Post-hoc test for target phalanges showed that there was significant (p < 0.001) decrease in smoothness when the position of the target phalanx was considered to change in the order from distal (−1.59) to proximal (−2.05). These findings are illustrated in the Fig. 8.

Figure 8 Change of smoothness of phalanx and thumb for different target phalanges.

The smoothness of the movement of digit (phalanx) and thumb was determined using spectral arc length method. When the spectral arc length value was large, lesser was the smoothness of the movement. The smoothness of the phalanx movement reduced significantly (p < 0.001) when the target position was changed from distal to proximal phalanx. There was no significant difference of smoothness of thumb movement when target positions were changed. The columns and bars indicate means and standard error of means.

Discussion

The special role of thumb in finger opposition

Traditionally, thumb has been considered to be a special finger. As stated by our colleagues Olafsdottir, Zatsiorsky & Latash (2005), thousands of years ago, life or death of defeated gladiators was decided by the thumbs-up or thumbs-down gesture. In today’s world, this gesture continues to hold crucial importance, although not on life or death decisions. For example, in the Internet, “like” or “dislike” of specific content is expressed by clicking the thumbs-up or thumbs-down buttons. In Indian mythology, great archer Ekalavya was tricked by his “assumed mentor” to amputate and offer his thumb as the fee for archery training to ensure that the mentor’s other (favorite) student would remain unchallenged.

Finger opposition movements form a fundamental component of various daily activities such as brushing, writing, shaping and force production in holding a cup of coffee without spilling etc. In our study, we investigated the movement amplitude and smoothness of thumb and finger phalanges while moving towards each other. Such an analysis would offer insights in to the mechanics and control of finger opposition movements. Two hypotheses were formulated as discussed in the introduction section. One of them was falsified in all cases while the second one was falsified in most cases. First Hypothesis was that the thumb and fingers would contribute approximately equally when trying to reach a distant target such as those in little finger but the contribution of thumb would be higher in reaching a nearer target (such as the proximal phalange of the index finger). This hypothesis was tested by introduction and development of an index called displacement ratio (D) for thumb and fingers. It should be noted that if this ratio for thumb increased, it means that thumb has contributed more when compared to fingers or vice-versa. Our results showed that the thumb contributes more even for distant targets (such as those in little finger), thus refuting this hypothesis. The second Hypothesis was that between the two effectors that reach for each other (say a finger phalanx and thumb), smoothness of movement would be higher for the effector that contributes more to a movement and lesser for the effector that contributes less. In almost all cases, it was found that the thumb contributes more to the movement. However, we found that in all cases the smoothness of the thumb was greater than the smoothness of the fingers. We discuss these results below.

Effects of Target fingers on DF and DT

Displacement ratio was calculated for fingers (DF) and thumb (DT) separately and were analyzed in the order of target fingers. The results showed that D for fingers (I, M, R, L) reduced significantly (p < 0.01) while the contribution of thumb increased significantly (p < 0.01), when the position of the target finger was considered to change in the order from index to little finger (0.34 (I) vs 0.65 (T) to 0.20 (L) to 0.79 (T)). This showed that once the opposition movement shifts from index to little, thumb makes a greater contribution when compared to its counterparts to execute the task successfully. A possible cause of this could be due to the musculature that is responsible for flexion of these digits. The muscles such as opponens pollicis, flexor pollicis brevis and flexor pollicis longus (Johanson, Valero-Cuevas & Hentz, 2001) produces flexion in thumb while muscles such as flexor digitorum profundus (FDP) and flexor digitorum superficialis (FDS) produces flexion of fingers (Fahrer, 1981; Von Schroeder, Botte & Gellman, 1990; Schuind et al., 1992; Leijnse et al., 1993; Leijnse, 1997). Note that both thumb and fingers need to move towards each other to achieve successful opposition.

It has been shown that muscles which produce flexion of fingers are structured as compartments that control flexion of different fingers such as Index, middle, ring and little by serving separate tendons to them (Fleckenstein et al., 1992; Schieber, 1995). These tendons that connect to the fingers are grouped close to the compartments in such a way where activating finger one may also activate other fingers (Kilbreath & Gandevia, 1994). However, the tendons that activate thumb are distinct from tendons serving other fingers. This is one of the reasons for the “lack of finger individuation” among digits. The thumb suffers relatively less due to this phenomenon when compared with the digits. In addition to peripheral architecture and morphology, possible overlap in neural substrates that control the fine activities of fingers and thumb is the other cause of this phenomenon (“Enslaving”, Zatsiorsky, Li & Latash, 1998; Zatsiorsky, Li & Latash, 2000). It is also known that the thumb has a disproportionately large representation within motor cortex (Penfield & Rasmussen, 1950), although, no clear boundary between thumb area and finger areas can be demarcated (Waberski et al., 2003). Hence, both central and peripheral factors probably contribute these phenomena (reviewed in Schieber & Santello, 2004).

Effects of Target phalanges on DF and DT

Further, by analyzing the displacement ratio in the order of target phalanges showed that the contribution of the thumb (T) increased while the contribution of fingers (F) reduced significantly (p < 0.001) when the position of the target phalanx was considered to change in the order from distal to proximal phalanx (0.45 (F) vs 0.54 (T) in distal phalanx to 0.11 (F) to 0.88 (T) in proximal phalanx). This suggests that the phalanx with a greater range of motion (Noort et al., 2016) appears to contribute more to the movement. Finger contribution reduces when the range of motion reduces. Therefore, it can be argued that distal phalanx is used along with thumb for fine dexterous movements (Napier, 1956; Signori et al., 2017) and the proximal phalanx is more frequently used in tasks requiring relatively less dexterity (and presumably more power), such that the phalanges are folded over the palm to produce power grip (Napier, 1956). Note that in both cases, displacement ratio (D) for the thumb is higher than the fingers, regardless of which finger or phalanx is the target. This suggests that the thumb makes greater contribution in all opposition movements. All these results were supported by 2-way repeated measures ANOVA and D was z-transformed before performing repeated measures ANOVA.

Effect of task level differences on DF and DT

Further, we analyzed changes of displacement ratio between tasks using one-way repeated measures ANOVA. The results from that analysis showed that when the targets were distal phalanges of index and middle fingers, D was statistically non-different between finger and the thumb (p > 0.05) suggesting that these targets made comparable contribution to finger opposition, although this was not seen in the ring finger and little finger, since, in all the other target cases, D for the thumb was greater (p < 0.001).

Effect of target fingers on movement smoothness of finger phalanx

Smoothness of finger movement was determined separately for fingers and were analyzed in the order of target fingers. The results showed that smoothness of finger phalanx significantly reduces (p < 0.01), when the position of the target finger was considered to change in the order from index to little finger (−1.76 (index) to −1.85 (little)). This appears to suggest that the data is in general agreement with our second hypothesis: the effectors that contribute more (in terms of displacement ratio), will also make smoother movements. Complex interconnections between the tendons of FDP and extensor digitorum communis in addition to connection within FDP have been reported earlier in literature (Kaplan, 1984; Von Schroeder, Botte & Gellman, 1990). It is possible that index and middle finger are better positioned to circumvent these complexities than ring and little fingers (Koshi, 2017). Such peripheral mechanical coupling has been traditionally considered to be a barrier to reaching higher skill levels and higher independence of finger movements that are required of some motor skills, such as those in arts (Leijnse et al., 1993; Leijnse, 1997).

Effect of target phalanges on movement smoothness of finger phalanx

Further, by analyzing smoothness of finger for different target phalanges, the smoothness of finger phalanx reduced significantly (p < 0.001) when the position of the target phalanx was considered to change in the order from distal (−1.59) to proximal (−2.05) phalanx. It is possible to argue that this performance of the distal phalanx may be purely due to superior morphology –the musculature and tendons. Another reason might be mainly because of the fact that the distal phalanx happens to be the tip in the serial manipulator chain (Zatsiorsky, 1998). Also, note that the distal phalanx is separately innervated with FDP (Kilbreath & Gandevia, 1994). Similarly, FDS innervates the middle phalanx resulting in better smoothness when compared with proximal phalanx which are only controlled by loose group of muscles, the lumbricals that attach on the tendons of FDS and distally to extensor group.

Thumb movement smoothness

The apparent advantage of superior morphology in producing smoother movements seems to persist in the case of the thumb. The thumb in humans has dedicated muscles, probably evolved from the multi-compartmental muscles of the fingers (Marzke, 1992) and probably specialized for opposition movements. Hence, thumb performance is superior to other fingers both in terms of relative contribution (displacement ratio D) and in terms of smoothness. Note that even in distal phalanx of the index and middle fingers, the only two targets where the relative displacement contributions of the thumb and fingers were statistically non-different, the thumb movement was smoother. This suggests that the second hypothesis is true in almost all cases but is probably more nuanced. In comparisons with the fingers, the thumb smoothness was always superior, regardless of relative displacement contribution.

Even though the thumb needs to move to different phalanges and fingers, there was no significant difference (p > 0.05) when the target position changed from index to little finger as well as from distal to proximal phalanx. This suggests that from the viewpoint of the thumb, opposition movements to all fingers and target positions are non-different in terms of movement smoothness.

Previous studies have also reported different cortical activations for index finger and thumb as well as for middle finger and thumb (Tanosaki et al., 2001; Järveläinen & Schürmann, 2002; Hamada et al., 2000). Such dedicated mechanical coupling at the periphery along with neural differences at cortical levels probably have given thumb an advantage in terms of better smoothness than other fingers and the ability to traverse between proximal phalanx of little to distal phalanx of index across the extremes of the hand with relative ease. This has probably resulted in the thumb being the major contributor in opposition movements resulting in successful day-to-day activities.

Concluding Comments

Finger opposition movement plays a critical role in performing daily activities. In the present study we have used displacement ratio and movement smoothness measures to suggest that the thumb produces larger movements that are also smoother when compared with the fingers. For fingers, displacement ratio and smoothness were more for index and middle when compared to ring and little. It is possible that this could be due to morphological and anatomical differences (biomechanics), which probably offers the simplest explanation of our results. However, more research is needed to confirm this. Overall, our results suggest that the thumb makes special and substantial contributions to opposition movements, the basis of most everyday dexterous manipulation.

Supplemental Information

Supplemental Information 1 Code used for data analysis and statistics

This zip file contains matlab and R scripts used to analyze data and perform statistics.

Click here for additional data file.

Supplemental Information 2 Movement segmentation algorithm

Movement segmentation algorithm described in detail with examples of how the algorithm segregates movements.

Click here for additional data file.

Additional Information and Declarations

Competing Interests

Author Contributions

Human Ethics

Data Availability

The authors declare there are no competing interests.

Dhanush Rachaveti conceived and designed the experiments, analyzed the data, prepared figures and/or tables, authored or reviewed drafts of the paper, approved the final draft.

Niranjan Chakrabhavi and Vaisakh Shankar conceived and designed the experiments, performed the experiments, authored or reviewed drafts of the paper, approved the final draft.

Varadhan SKM conceived and designed the experiments, contributed reagents/materials/analysis tools, authored or reviewed drafts of the paper, approved the final draft, acquired funding for the study.

The following information was supplied relating to ethical approvals (i.e., approving body and any reference numbers):

All experimental procedures were approved by the institutional ethics committee of Indian Institute of Technology Madras (IEC/2018/01/SKM-2/03).

The following information was supplied regarding data availability:

Rachaveti, Dhanush; Chakrabhavi, Niranjan; Shankar, Vaisakh; SKM, Varadhan (2018): Thumbs up: Movements made by the thumb are smoother and larger than fingers in finger-thumb opposition tasks figshare. Dataset. https://doi.org/10.6084/m9.figshare.5945578.v1.

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
