# Peer review of "Thumbs up: movements made by the thumb are smoother and larger than fingers in finger-thumb opposition tasks"

_PeerJ, doi:10.7717/peerj.5763_

## Round 0.1 · original submission · Major Revisions

The two reviewers and I feel that your manuscript has promise and therefore would like to offer you the chance to respond to our comments.

·

Basic reporting

The English needs significant editing.

As the study focuses on movements of the fingers, it is strange that there is not a single graph showing mean or representative kinematic data. It is essential that this data be included to allow the readers and reviewers to evaluate the claims made in the paper.

The motivation of the task is not clear - opposition of distal phalanges is commonly used in day-to-day life (albeit usually with an object in between), while opposition of the thumb with the other phalanges is much less frequent. Can you give examples from day-to-day life when these movements are used?

Experimental design

The sample size (N=8) seems small, and no sample-size analysis was performed. Please justify why this sample size was selected.

The measure E_F is redundant- it always equals 1-E_T. So there is no need to analyze it separately. When you want to check if E_F and E_T are different, just compare E_F to 0.5

The smoothness measure is unclear - are you measuring smoothness of a particular phalange, or the whole finger? joint angles or position?

The methods are unclear, I do not feel that I could replicate the calculations:
- Is any filtering performed? Calculation of velocities without filtering is very problematic
- In the calculation of R_D and R_V, it is not clear what is the base of the coordinate system. Are separate calculations performed for the thumb and finger? which phalange? How is movement onset and end determined (it is written based on maxima and minima, but how are these defined?)
- The specteral arc length (SAL) used is difficult to understand intuitively - what should be considered a meaningful difference in smoothness in terms of SAL?

Validity of the findings

Due to the problems described above, I found it difficult to determine the validity of the findings.

Regarding the smoothness, I would like to see comparisons of the velocity profiles of the finger and the thumb (graphs) so I can judge whether one looks more "smooth" than the other. Having different smoothing from the two effectors performing a coordinated action seems sub-optimal.

Regarding the exertion index, one major factor has been ignored, namely the range of motion of the fingers. The biomechanics of the hand do not allow arbitrary movements of the fingers. Hence the differences observed in "exertion" between the thumb and fingers made be purely due to biomechanical constraints. I would like to see this taken into account in some way (e.g. by comparing movements to the edges of the range of motion of the finger joints)

Additional comments

Minor corrections:

line 51: phalanx positions is mentioned here in abstract with no previous reference
line 65: this claim seems hard to believe
line 73: forces coordinate with each other -> forces are coordinated
line 80/81: shape, contour of the hand. This is unclear what you mean
line 99: has -> have
line 100: healthy from unhealthy. these terms are not precise
line 110: The first hypothesis
line 113: The second hypothesis
line 117: This hypothesis needs justification, it is not obvious why this is true
line 134: An electromagnetic
line 134: Describe which system, not just the company (Liberty?)
line 152: at a comfortable speed
line 152: how were they instructed? verbally? on the screen? did they have to start moving in response to a beep?
line 156: delete for
line 160: The experimenter monitored
line 163: this sentence is redundant (was explained previously)
line 171: I have difficulty understanding what R_D is. From your equation, it seems like it is the distance from the origin of the coordinate system (the transmitter). This seems like a strange choice. Shouldn't it be the distance between the fingers, or the distance from the palm of the hand?
line 174: R_D has a bell-shaped profile - this is not obvious why - later you investigate smoothness, if all movements are bell-shaped they presumedly have similar smoothness values
line 192: The reference to your previous studies for calculating a standard measure (the z-score) seems self-serving
line 213: I assume you mean H-F was used when sphericity was violated\
line 215: This paragraph is redundant, no need to redefine the measure
line 227 and elsewhere: units are missing!
line 250: Why 12 one-way ANOVAs and not a single repeated measures? Running this number of ANOVAs without corrections is problematic
line 257: redundant paragraph
line 278: "wonderfully" - not scientific language
line 316: The thumb
line 326: This is not an explanation, rather just states a fact
line 332: What do you mean by flex? Some joints must flex, but not all joints.
line 375: anatomical architecture - this a vague term, it is not clear what is meant

Reviewer 2 ·

Basic reporting

This is a generally well written and presented manuscript. The authors make appropriate references to the literature.

At the moment the raw data for the work is not available.

There are two hypotheses, the first hypothesis has two parts but there is no rationale for the second part of the hypothesis.

Experimental design

The experimental design is good.

There are three primary issues with the Methods,
1) Reference is made in to segment angular orientations, but these are never used. Reference to the angular orientations should be removed.

2) Velocities are analyzed, but no information is provided on data filtering and differentiation.

3) The authors present a metric, the "Exertion Ratio", but what is quantified only indirectly related to exertion. This ratio should be more appropriately named, for example the "Movement Ratio".

Validity of the findings

No Comment.

Additional comments

Line 33 - delete second "the".
Line 34 - "...activities of the hand.".
Line 37 - "...has not been well understood.".
Line 49 - "The Exertion Ratio...".
Line 51 - better in what sense?
Line 55 - "...made a greater...".
Line 61 - "...of the opposite...".
Line 67 - delete "to".
Line 78 - replace "this" with "that".
Line 101 - please provided references for these comparisons of novices and experts.
Line 108 - "...of the thumb.".
Line 160 - "An experimenter monitored...".
Line 201 - this in text citation should be "...et al. (2012)...". This should be corrected elsewhere in the manuscript.
Lines 216-221 - this is not a results but details which should be in the Methods section.
Line 316 - "The thumb showed...".
Reference 40 - this reference has a different format to the other references.

---

## Round 0.2 · Minor Revisions

The two reviewers and I are happy with your attempt at responding to our initial concerns. However I suggest you look at the remaining concerns of the two reviewers and work diligently to take on board the comments if you wish your paper be accepted for publication.

·

Basic reporting

The authors have improved the presentation of the data by adding the graphs and more details.

The presentation of the graphs is fine, but it would be more useful to combined figures 1 & 2, and figures 3 & 4 (e.g. used dashed line for thumb). In this way, it makes it easy to observe the claims made in the paper (regarding the relative size of the movements, and the differences in velocity profiles).

I still have a problem with the equation for R_D. The current equation implies that you take the distance relative to the base of the hand coordinate system. I assume you actually take the displacement from the start of the movement. So the equation should be something like: R_D = sqrt( (x - x_0)^2 + (y-y_0)^2 + (z-z_0)^2))

Also, in the equations for D_T and D_F, please use the notation of R_D you defined above.
The authors have improved the presentation of the data by adding the graphs and more details.

The presentation of the graphs is fine, but it would be more useful to combined figures 1 & 2, and figures 3 & 4 (e.g. used dashed line for thumb). In this way, it makes it easy to observe the claims made in the paper (regarding the relative size of the movements, and the differences in velocity profiles).

I still have a problem with the equation for R_D. The current equation implies that you take the distance relative to the base of the hand coordinate system. I assume you actually take the displacement from the start of the movement. So the equation should be something like: R_D = sqrt( (x - x_0)^2 + (y-y_0)^2 + (z-z_0)^2))

Also, in the equations for D_T and D_F, please use the notation of R_D you defined above.

Experimental design

no comment

Validity of the findings

no comment

Additional comments

I found the comment about traditional south Indian music / counting interesting, and a good motivation for the study, I would add these lines to the introduction.

Regarding speculation on the differences in the exertion index - in the discussion when you bring this topic up, I think you should at least mention that the differences observed may be due to the biomechanics of the hand. Although you did not measure range of motion of the fingers, this data is readily available, and seems to present the most parsimonious explanation for your findings.

Reviewer 2 ·

Basic reporting

The revised manuscript is much better than the previous version in all of these aspects.

Experimental design

The manuscript is good in these aspects, and with use now of the term displacement index is superior to the term term used in previous submission.

Validity of the findings

The research provides some new insights into the function of the thumb.

Additional comments

Specific Comments (line numbers refer to the version with tracked changes)
Line 40, Comment – change “has not been well understood” to “ is not well understood”.
Line 91, Comment – in what sense special?
Line 120, Comment – “…of the thumb.”.
Line 140, Comment – “…using the Edinburgh…”.
Line 187, Comment – “…only onward movements were analyzed.”.
Line 195, Comment – reference needed for stencil.
Lines 215-217, Comment – the process outlined here does not find the first time when 5% of the maximum velocity was crossed.
Line 402, Comment – “e.g.” means in Latin exempli gratia, which means “for example” therefore there is redundancy in this sentence.
Line 420, Comment – “The second hypothesis…”.
Line 521, Comment – “…smoothness of finger movement…”.
Line 524, Comment – here and elsewhere what do you mean by “superior”?
Line 534, Comment – “…case of the thumb.”.
Line 537, Comment – “…in terms of movement smoothness.”.
Line 556, Comment – not “day-to-day” to “daily”.
Line 559, Comment – “…compared with the fingers.”.

---

## Round 0.3 · accepted · Accept

We thank you for the hard work you put in to attending to the comments of the reviewers and I. We are pleased to let you now that your paper has been accepted for publication in PeerJ,

# ·

Basic reporting

I am satisfied with the changes made by the authors. In particular, combining the graphs makes it much easier to understand the central claims of the paper.

Experimental design

no comment

Validity of the findings

no comment

Reviewer 2 ·

Basic reporting

Acceptable.

Experimental design

Acceptable.

Validity of the findings

Acceptable.

Additional comments

The manuscript is much improved in this submission. I have no further comments.